# Wine *Saccharomyces* Yeasts for Beer Fermentation

Vanesa Postigo [1,2], Margarita García [1], Juan Mariano Cabellos [1] and Teresa Arroyo [1,*]

[1] Department of Agri-Food, Madrid Institute for Rural, Food and Agriculture Research and Development (IMIDRA), El Encín, A-2, km 38.2, 28805 Alcalá de Henares, Spain; vanesa.postigo@madrid.org (V.P.); margarita_garcia_garcia@madrid.org (M.G.); juan.cabellos@madrid.org (J.M.C.)

[2] Brewery La Cibeles, Petróleo 34, 28918 Leganés, Spain

[*] Correspondence: teresa.arroyo@madrid.org

**Abstract:** Multiple studies in recent years have shown the potential of *Saccharomyces* wild yeasts to produce craft beers with new flavour profiles and other desirable properties. Yeasts isolated from food (wine, bread, kombucha . . . ) have shown potential promise for application in brewing. The aim of this study is to evaluate the ability of 141 *Saccharomyces* yeast strains isolated from the Madrilenian agriculture (from grapes, must, wine, vineyard, and cellars) to produce a novel ale beer. Fermentation activity of the strains was compared against the commercial strain *Saccharomyces cerevisiae* Safale S-04. In addition to the other aspects such as melatonin production, thirty-three volatile compounds belonging to higher alcohols, esters, aldehydes/cetones, acids, lactones and phenolic groups, were analysed by GC for selection of the strains. Ten strains were finally chosen, among which the most relevant was the strain G 520 showing a higher production of esters, higher alcohols and acids compared with S-04. The apparent attenuation for this strain was lower than commercial strain, which translates into more residual sugars. Furthermore, G 520 was more capable of producing significantly higher amounts of melatonin studied by HPLC, as well as showing a higher antioxidant capacity. Consumer study showed that G 520 strain could be used to produce a potential beer that has a place in the current market.

**Keywords:** beer; wild *Saccharomyces cerevisiae*; volatile compounds; functional beer

## 1. Introduction

The craft beer industry has increased during recent years in Europe, with a growing consumer interest in new beer styles and new flavours [1]. Beer production is divided into different styles, with ale and lager being the most predominant. Lager beers, also known as bottom or low, are carried out at 6–15 °C. In contrast, ale beers are produced by top-fermenting yeasts at fermentation temperatures between 16 and 24 °C. At the end of fermentation, top yeast tends to rise to the surface of the fermented wort, forming a thick film, whereas bottom yeast settles to the bottom of the fermentation vessel. Furthermore, ales are known for their fruity aromas, while lager beers are more neutral [2]. To produce innovative specialty ale beers with high added value, brewers have used a variety of ingredients, with special interest in yeast, the main objectives being to improve the quality of the beer, increase the efficiency of fermentation, and to provide characteristic sensory notes. Therefore, the choice of yeast is an important factor to achieve a product that is valued by consumers and that has the required distinctive features and flavours. Several studies have isolated *Saccharomyces* yeast strains from different foods, as they could produce distinctive fermentative aroma profiles in beer [3–5].

The yeasts not only metabolise the sugar content in the wort into $CO_2$ and ethanol but are also responsible for the production of several by-products during fermentation, including higher alcohols, esters, aldehydes, vicinal diketones (VDKs), $H_2S$ and acids [6,7]. While higher alcohols and esters are desirable volatile components for a pleasant beer, VDKs

and $H_2S$ are often considered as off-flavours which are critical issues in beer production. Although the concentrations of esters are low, they are considered to be the most important contributors to beer flavours, and more than 90 different esters can be produced in beer [6,8]. Some acids such as hexanoic acid, octanoic acid and decanoic acid are responsible for off-flavours such as caprylic flavour, considered as an unpleasant flavour and therefore unwanted in beer [9,10]. The development of off-flavours in beers is a handicap for brewers, as they generally do not use pasteurisation and/or filtering processes for microbiological stabilisation [11].

Compared with conventional brewing yeast, such species (isolated from different foods or environments) offer several functional benefits, not only regarding the aromas production, but also for the production of compounds such as melatonin and compounds with antioxidant activity [12,13].

The objective of this study was to evaluate the fermentative capacity of 141 *Saccharomyces* yeast strains isolated from Madrilenian agriculture and their ability to produce potentially functional beers. For this purpose, the selected strains were analysed for the main parameters in beer (residual fermentable sugars, ethanol content, glycerol, colour, bitterness, lactic acid, $SO_2$ and VDKs), volatile profiles, sensory analysis (trained panel and consumers), melatonin production and antioxidant activity, to produce a novel beer.

## 2. Materials and Methods

### 2.1. Yeast Strains

The 141 yeast strains used in this study belong to the Autochthonous Yeast Collection of the Madrid Institute for Rural, Food and Agriculture Research and Development (IMIDRA, Madrid, Spain). These had been isolated from different environments (from grapes, must, wine, vineyard and cellars) related to D.O. "Vinos de Madrid". They are preserved under cryogenization at $-80\,°C$ (YPD broth supplemented with 40% ($w/v$) glycerol). The *Saccharomyces cerevisiae* commercial strain S-04 (Fermentis, Lesaffre, Marcq-en-Barœul France) was used as the control during the fermentation trials.

To ensure the identity and purity of all the strains, they were tested by microsatellite multiplex PCR, using the highly polymorphic loci SC8132X, YOR267C and SCPTSY7 [14].

### 2.2. Fermentation Screening

A malted barley wort produced in La Cibeles (Madrid) brewing plant was used for the fermentation experiments to elaborate the ale-style beer. All beers were made following the same recipe and the mean analytical characters of wort were as follows: pH 5.7; gravity, 11.55° Plato and 1.047 g cm$^{-3}$; free amino nitrogen, 235.65 ppm, bitterness, 32.77 IBU.

The fermentations were carried out by triplicate using inoculums grown in YPD broth medium (1% bacteriological peptone, 2% yeast extract and 2% glucose; all $w/v$) (Condalab, Madrid, Spain) at 28 °C for 24 h.

#### 2.2.1. Lab Scale Fermentations

The fermentation capacity of the 141 yeast strains studied were firstly tested under anaerobic conditions at 20 °C and with rotary shaking (120 rpm) in 150 mL bottles containing 100 mL of sterilised wort. Fermentation progress was monitored daily by weight determination until constant weight. Once the fermentation is finished, samples were centrifuged and stored at $-30\,°C$ until their analysis.

The ability to ferment maltose using Durham test [15] and the production level of $H_2S$ was studied. The yeasts were grown on bismuth-containing indicator medium BIGGY agar (Oxoid) plates at 28 °C for 2 days [16]. A colour scale from 1 to 4 points was built based on the browning of yeast colonies in the medium BIGGY agar due to their $H_2S$ production:

- Type I (white/creme): low-null production $H_2S$
- Type II (light brown): moderate production $H_2S$
- Type III (brown): high production $H_2S$
- Type IV (dark brown/black): very high production $H_2S$

In a second phase, based on the results obtained in relation to the use of maltose, sulphidric production and ability to ferment beer wort, 126 yeast strains were selected and studied in 1 L fermenters containing 900 mL of sterilised wort, and locked with a Müller valve containing sulphuric acid, to allow the $CO_2$ to escape from the system. Fermentation conditions were the same as previous: 18 °C, stirring 120 rpm and automatic control of lost weight each hour. After the fermentation is finished, beers were added with 7 g $L^{-1}$ of glucose (sterilized under ultraviolet light), bottled, and stored at 20 °C during one week for bottled conditioning and one month at 4 °C for maturation. Yeast viability before bottling was also measured. The resultant beers were sensorially and analytically analysed.

### 2.2.2. Brewery Scale Fermentations

Selected yeast strains were tested at the brewery. The fermentations were performed at 18–20 °C in 100 L fermenters containing 90 L of wort, under static conditions and same recipe. Pre-cultures were grown in wort at different sequential scales for five days at 20 °C. Fermentation activity was measured by density every day until it was established (constant density for three consecutive days). Maturation was carried out at 0–4 °C for 3–4 weeks and then under a $CO_2$ pressure for five days. Bottling was done using manual isobaric equipment under anti-oxidative conditions.

### 2.3. Analytical Methods

After maturation, the 1 L and 100 L beers from the scaling process were analysed for different parameters. The samples were degassed through a cellulose filter of grade 2 V (Whatman, Maidstone, UK) prior to analysis.

Colour (range 1–100 EBC- European Brewing Convention), bitterness (range 5–100 IBU—International Bitterness Unit), lactic acid (range 150–3500 ppm), vicinal diketones: diacetyl and 2-3-pentanodione (VDKs) (range 0.05–2 ppm) and $SO_2$ (range 1–30 ppm) were analysed with CDR FoodLab (BeerLab software, www.cdrfoodlab.com (accessed on 8 November 2021), verified by the international reference analysis laboratory Campden BRI (www.campdenbri.co.uk, Compare products section, CDR BeerLab Touch Analyser, accessed on 29 November 2021).

### 2.3.1. Residual Sugars, Glycerol, and Ethanol Determination

The analysis of residual fermentable sugars: maltotriose, maltose, glucose and fructose, glycerol and ethanol were performed on a HPLC equipment Dionex Ultimate 3000 (Thermo Scientific, Waltham, MA, USA), equipped with a quaternary pump, an autosampler, a column compartment provided with a temperature controller and a 520 refractive index detector (ERC). Separations were carried out on an ion exclusion column, Rezex ROA-Organic Acid H+ (8%), 150 × 7.8 mm (Phenomenex) using water with hydrogen sulphide (0.005 N, HPLC grade, Panreac, Barcelona, Spain) as mobile phase at 0.6 mL $min^{-1}$. The injection volume was 10 μL and the oven temperature 60 °C [17]. Samples were previously filtered through a 0.22 μm filter. All calibrations produced a linear response with an $R^2$ value > 0.9881 over the concentration range analysed.

### 2.3.2. Volatile Compounds by GC

Higher alcohols, esters, acids, acetaldehydes-cetones, lactones and phenols (33 major aromatic compounds) were determined according to method from Ortega et al. [18] based on liquid-phase microextraction with dichloromethane (DCM) and then identified with gas chromatography. A gas chromatograph 6850 (GC-FID, Agilent Technologies, Inc., Santa Clara, CA, USA) equipped with a flame ionization detector was utilized for analysis. The column used was a DB-WAX (60 m × 0.32 mm i.d. and 0.5 μm film). Helium was used as a carrier gas at a flow rate of 2 mL $min^{-1}$, with oven temperature: 5 min at 40 °C, followed by 3 °C $min^{-1}$ increase up to 200 °C; injector and detector temperatures: 200 °C and splitless injection.

Four internal standards (2-butanol, 4-methyl-2-pentanol, 4-hydroxy-4-methyl-2-pentanone, 2-octanol) were used for determination of major aromatic compounds. Levels of the different compounds were determined by calibration lines for each compound ($R^2$ = 0.9861–0.9969).

### 2.3.3. Melatonin Production by HPLC

Melatonin was extracted using solid phase extraction (SPE) with RP-18 standard PP-tubes (Agilent Technologies, Inc., Santa Clara, CA, USA). The extraction columns were prepared and conditioned with 2 mL of methanol (Scharlab, Barcelona, Spain) and 5 mL of bidistilled water, loaded with 500 µL of beer, impurities washed with 2 mL bidistilled water and eluted with 2 mL of methanol [19]. The eluted were dried inside a thermoblock at 80 °C under a nitrogen stream. Finally, the dry extracts were reconstituted with 300 µL of methanol and 700 µL of mobile phase (formic acid (0.1%)/Acetonitrile (95:5)) (HPLC grade; Carlo Erba, Italia/Panreac-Applichem, Barcelona, Spain). The reconstituted extract was filtered through a 0.22 µm syringe filter (13 mm hydrophilic PVDF syringe filter) before 10 µL volume injection in HPLC system.

Melatonin content was determined by HPLC using a Waters 600 HPLC controller system connected to an autosampler Waters 717 plus and a Waters 2475 multifluorescence detector. The fluorescence detector recorded wavelengths of 270 nm for excitation and 372 nm for emission. The chromatographic separations were performed on a ZORBAX Eclipse Plus C18 column (Agilent Technologies, Inc., Santa Clara, CA, USA) using the mixture of 0.1% formic acid in water and acetonitrile (95:5) at a flow rate of 1 mL min$^{-1}$ at 30 °C. The concentration of melatonin was measured by using a linear calibration curve ($R^2$ > 0.9856) [20–22].

### 2.3.4. Determination of Antioxidant Activity

The antioxidant activity was determined in beers with the e-BQC lab device (Bioquochem, Asturias, Spain, www.bioquochem.com, accessed 8 November 2021), based on a redox potential measure and expressed in micro-Coulombs (µC) [23]. Two values were measured: the antioxidant capacity of the compounds with the highest rate of free radical scavenging (Q1) and with a lower rate of free radical scavenging (Q2).

The determination of antioxidant capacity was calculated by TEAC (Trolox Equivalent Antioxidant Capacity) assay using a solution of 6-hydroxy-2,5,7,8-tetrametilchroman-2-carboxylic acid (Trolox 8 mM L$^{-1}$ in methanol 5% and pH 4.5). Using the Trolox calibration curve (Q1, $R^2$ = 0.9974; Q2, $R^2$ = 0.9876; QT = Q1 + Q2), expressed as e-BQC measurement versus concentration (µmol L$^{-1}$), the antioxidant activity of the beers was expressed as millimoles of Trolox equivalents per litre (mmol TE L$^{-1}$).

### 2.4. Sensory Analysis

A panel of ten trained and experienced beer tasters (five male, five female) evaluated different attributes in 1 L and 100 L beers. The panel was trained in various beer flavours (diacetyl, DMS, acetaldehyde, bitter, butyric acid, isovaleric acid, lactic acid, earthy, $H_2S$, geraniol, clove, grainy, papery, indole, light-struck) (Siebel Institute of Technology, Chicago, Illinois) according to EBC method 13 [24–26]. A standardised tasting room with individual booths equipped with a glass of water and with room temperature of 24 °C to 25 °C was used.

This test performs a formal and structured quantitative descriptive analysis to obtain a flavour profile of the beers, assessing the intensity of each attribute. The overall quality scores are also used to rate the overall quality of the beers (EBC method 13.10) [27]. The attributes were classified into three groups: appearance (colour, foam retention), smell (esters, alcohols . . . ) and taste (alcohol, sweet, salty, acidic, bitterness, astringency, effervescence, warmth, slickness, body). Each attribute was rated from absence to presence using a scale from 0 to 5 points. A radar chart was elaborated showing the main average values of different attributes for each beer.

### 2.5. Consumer Acceptability Test

A sensory test with 112 consumers (63.72% females and 36.28% males, with an average age ranged from 20 to 73 years old) was conducted to assess aroma, flavour and overall liking of the beer sample. A 9-point hedonic scale for categorising bitterness, acidity, and overall impression was used, in which 1 = dislike extremely, 5 = neither like nor dislike and 9 = like extremely. On the other hand, the rest of the attributes that could be found in beer were analysed according to Check-all-that-apply (CATA) questions [28], which allows multiple options to be selected. The session was conducted in a normalized sensory room from IMIDRA in individual booths using the Sensesbit software (TasteLab, Galicia, Spain, www.sensesbit.com, accessed 8 November 2021) to display the questionnaire. The amount of sample provided to each participant was 50 mL of beer at 10 °C.

### 2.6. Statistical Analysis

Data from chemical and volatile compounds of the beers elaborated in 1 L fermenters were analysed statistically and presented in means and standard deviation. One-way analysis of variance (ANOVA) with Tukey post-hoc test on a significance level of $p < 0.05$ was done for the different strains. Identification of significant data correlation was performed with the Pearson test. Principal component analysis (PCA) was done to visualize the beer samples in a n-dimensional space by identifying the direction in which most of the information were retained. This shows the separation between the different yeast strains and the main components they target (total higher alcohols, total esters, total aldehydes/cetones, total fatty acids, γ-butyrolactone, guaiacol, bitterness, glycerol, ethanol, lactic acid, colour, melatonin and antioxidant capacity) and thus allows to determine whether strains isolated in Madrilenian agriculture tend more towards one or the other component or are not related to each other. The program used was R Studio 4.1 (Integrated Development for R. RStudio, PBC, Boston, MA, USA).

## 3. Results and Discussion

### 3.1. Screening of S. cerevisiae Strains at Lab Scale

The choice of a yeast strain to use in the brewing process is fundamental to achieve a valued product with distinctive characteristics and flavours demanded by consumers [29]. The evaluation of 141 *S. cerevisiae* strains was carried out to determine their fermentation performances for beer production. To achieve this objective, it was necessary to study the ability to ferment maltose, the production level of $H_2S$ and their fermentative behaviour in wort:

- The content of fermentable sugars in wort depends on the raw ingredients (mostly barley malt) and the method used in wort preparation. The order in which yeasts metabolise the fermentable sugars is as follows, glucose, fructose, maltose and maltotriose, maltose being the most abundant in wort. Complete and timely conversion of all sugars by *Saccharomyces* strains is the purpose for brewers. In this context, not all species are able to consume the four sugars [4]. The order in which sugars are fermented, may be the answer for non-maltose fermenting strains. Some studies suggest that glucose could control the maltose metabolism, repressing the synthesis of maltose transporters and of the α-glucosidases (maltases) that hydrolyse this sugar inside the cell [30,31]. Furthermore, maltose transport is more strongly inhibited by glucose in some ale strains than in some lager strains [32,33]. This could be an explanation for the behaviour of twelve yeast strains found in the study that were not able to ferment maltose.

- A total 70.9% of the strains showed a moderate production of $H_2S$ (Type II), whereas 10.3% showed a high production, including commercial strain S-04 (Type III) and 18.8% a low-null production. $H_2S$ is a volatile compound mostly unwanted, as it is responsible for a "rotten-egg" smell, thus masking other desired aromas in beer [34]. Its concentration changes during the fermentation process due to the depletion of fermentable sugars, with a rapid decrease observed when the assimilation rate falls

below 0.05 $w/w$% h$^{-1}$. However it could also vary by yeast capture of H$_2$S at the end of the fermentation in green beer [35]. For this reason, despite the fact that most of the yeast strains were within production Type II, these aromas were not found in the final beers.

- The yeast strains were able to ferment wort in 100 mL fermentation, but in lower levels than the *Saccharomyces* control strain, S-04.

Based on the screening results (ability to ferment maltose, H$_2$S production) and by smelling the 100 mL matured beer, 126 strains were preselected to scale them up in 1 L. The 15 yeast strains that were discarded showed off aromas not desired in beer in some styles such as diacetyl, sulphur-containing compounds, phenolic off-flavours (POFs) (beers with a strong medicinal, clove-like aroma), but essential flavour in Belgian white beers, German Weizen and Rauch beers [36,37]. It is also worth mentioning the selection of three yeast strains (G 487, CLI 275, CLI 1056) that, despite being within type III in terms of H$_2$S production, like strain S-04, did not present undesired aromas and it was decided to test them in the following scale-up to check that this behaviour was still maintained.

The 126 yeast strains were tested in 1 L fermenters with a fermentation kinetics of approximately 7–10 days, depending on the strain and yeast viable cells ranging between 0.4 and $2.8 \times 10^7$ mL$^{-1}$ before bottling. Once beer was matured, they were tasted and those with no undesirable aromas and a good balance between aroma and taste were selected (ten strains) and analysed for different parameters.

Table 1 shows the data for the fermentation kinetics and residual fermentable sugars of the resulting beers fermented with the ten selected strains in 1 L. The S-04 commercial strains showed the best performance regarding apparent attenuation (82.2%), but the strains of the study showed good performance as well (between 68.9 and 72%) and an ethanol production between 4.43 and 5.10% $v/v$. Only two strains differ from these results, G 4 that was not able to ferment maltose in wort and therefore had very low ethanol production (1.59% $v/v$), but it was selected because of its aromatic profile, and the yeast strain CLI 1056 that fermented glucose partially with an apparent attenuation of 64% but it produces suitable ethanol levels (3.97% $v/v$). The commercial strain consumed 96% of the sugars quantified (including trisaccharides, disaccharides and monosaccharides), while the strains studied did not ferment maltotriose and all the maltose available in wort. From a sensory point of view, the incomplete fermentation of maltose and maltotriose, as is the case for the strain G 4, could give rise to sweet beers [38]. In addition to contributing to the sweetness of the beer, residual sugars can influence physical properties such as viscosity, which could contribute to the body or mouthfeel [39,40]. The use of alternative, maltose-negative yeasts is a useful way to produce low-alcohol beers that exhibit aromatic complexity. Such yeasts also reduce the aldehydes in the wort, thus eliminating the "worty" taste often found in low-alcohol beers [38,41]

Glycerol is a major by-product of ethanol fermentation by *Saccharomyces cerevisiae* [42]. This compound contributes to body and mouthfeel and it influences beer flavour [43,44]. Its production is closely connected to the growth rate of the cells, ethanol production and sugars utilization. Glycerol levels were found between 3.1 and 3.8 g L$^{-1}$ while for the G4 strain it was lower (1.75 g L$^{-1}$). The fermentation parameters showed significant correlation of ethanol concentrations and glycerol in all yeast strains (r < 0.87, $p$ = 0.01).

### 3.2. Beer Analysis

The main analytical characteristics of the resulting beers were analysed with CDR FoodLab and are shown in Table 2.

**Table 1.** Fermentation kinetics parameters and residual sugars of the ten selected strains in 1 L fermentation and the S-04 strain.

| Yeast Strains | $CO_2$ (g $L^{-1}$) | Residual Fermentable Sugars (g $L^{-1}$) | | | | Apparent Attenuation (%) | Ethanol % (v/v) | Glycerol (g $L^{-1}$) |
|---|---|---|---|---|---|---|---|---|
| | | Maltotriose | Maltose | Glucose | Fructose | | | |
| G 4 | 12.35 ± 1.06 [f] | 13.68 ± 0.04 [a] | 63.18 ± 0.02 [a] | 0.94 ± 0.01 [b] | 0.31 ± 0.00 [a] | 21.00 ± 0.01 [f] | 1.59 ± 0.02 [e] | 1.75 ± 0.01 [e] |
| G 354 | 51.55 ± 3.32 [cd] | 13.33 ± 2.24 [abc] | 3.75 ± 0.03 [b] | 0.70 ± 0.03 [c] | 0.17 ± 0.01 [b] | 72.00 ± 0.00 [b] | 5.07 ± 0.04 [b] | 3.63 ± 0.04 [ab] |
| G 450 | 51.05 ± 2.33 [cd] | 12.81 ± 0.08 [cde] | 3.40 ± 0.08 [bc] | 0.57 ± 0.00 [cd] | 0.17 ± 0.00 [b] | 68.90 ± 0.00 [d] | 4.63 ± 0.02 [c] | 3.10 ± 0.08 [d] |
| G 487 | 51.70 ± 0.14 [cd] | 12.48 ± 0.06 [e] | 3.45 ± 0.03 [bc] | 0.62 ± 0.01 [c] | 0.16 ± 0.00 [b] | 68.90 ± 0.00 [d] | 4.55 ± 0.02 [c] | 3.58 ± 0.15 [abc] |
| G 502 | 57.95 ± 0.35 [b] | 13.32 ± 0.01 [abcd] | 3.53 ± 0.03 [bc] | 0.66 ± 0.01 [c] | 0.18 ± 0.00 [b] | 72.00 ± 0.00 [b] | 5.10 ± 0.04 [b] | 3.68 ± 0.08 [a] |
| G 520 | 47.70 ± 0.99 [de] | 13.24 ± 0.01 [abcd] | 3.30 ± 0.00 [cd] | 0.57 ± 0.01 [cd] | 0.20 ± 0.01 [b] | 72.00 ± 0.00 [b] | 5.07 ± 0.05 [b] | 3.19 ± 0.14 [cd] |
| CLI 70 | 48.50 ±1.41 [de] | 12.73 ± 0.56 [cde] | 3.14 ± 0.38 [cd] | 0.40 ± 0.23 [d] | 0.19 ± 0.02 [b] | 68.90 ± 0.00 [d] | 4.43 ± 0.14 [c] | 3.26 ± 0.07 [bcd] |
| CLI 275 | 46.55 ± 0.49 [de] | 12.97 ± 0.01 [bcde] | 2.98 ± 0.03 [d] | 0.61 ± 0.01 [c] | 0.18 ± 0.00 [b] | 71.10 ± 0.03 [c] | 4.56 ± 0.01 [c] | 3.25 ± 0.02 [bcd] |
| CLI 1056 | 44.45 ± 2.47 [e] | 12.70 ± 0.31 [cde] | 3.52 ±0.24 [bc] | 5.72 ± 0.05 [a] | 0.15 ± 0.06 [b] | 64.00 ± 0.00 [e] | 3.97 ± 0.16 [d] | 3.55 ± 0.35 [abc] |
| CLI 1109 | 55.50 ± 2.83 [bc] | 13.50 ± 0.12 [ab] | 3.43 ± 0.04 [bc] | 0.6 ± 0.00 [c] | 0.19 ± 0.01 [b] | 71.00 ± 0.01 [c] | 5.07 ± 0.03 [b] | 3.61 ± 0.10 [ab] |
| S-04 | 66.5 ± 0.85 [a] | 1.43 ± 0.08 [f] | 1.45 ± 0.01 [e] | 0.16 ± 0.00 [e] | 0.18 ± 0.02 [b] | 82.22 ± 0.00 [a] | 5.53 ± 0.05 [a] | 3.43 ± 0.01 [abcd] |

Data are means ± standard deviations of three independent samples. $CO_2$ evolved until the fermentation is stable. Data with different superscript letters within each column are significantly different (Tukey tests: $p < 0.05$). Initial fermentable sugars, maltotriose: 13.98 ± 1.63 g $L^{-1}$, maltose: 58.66 ± 3.74 g $L^{-1}$, glucose: 12.53 ± 4.59 g $L^{-1}$, fructose: 2.26 ± 0.17 g $L^{-1}$.

**Table 2.** Main analytical characteristics of ten selected strains and S-04 in 1 L fermentation analysed with CDR FoodLab.

| Yeast Strains | Lactic Acid ppm | Colour EBC | Bitterness IBU | VDKs ppm | $SO_2$ ppm |
|---|---|---|---|---|---|
| G 4 | 208.50 ± 3.50 [c] | 12.00 ± 0.00 | 34.40 ± 0.30 [a] | ≤0.05 | ≤1 |
| G 354 | 278.50 ± 21.50 [ab] | 11.00 ± 0.00 | 30.60 ± 2.80 [ab] | ≤0.05 | 1.90 ± 0.00 [b] |
| G 450 | 266.00 ± 7.00 [abc] | 11.00 ± 0.00 | 28.60 ± 2.00 [abc] | ≤0.05 | 2.05 ± 1.05 [b] |
| G 487 | 417.50 ± 12.50 [a] | 11.00 ± 0.00 | 32.30 ± 2.00 [a] | ≤0.05 | 1.70 ± 0.70 [b] |
| G 502 | 236.00 ± 13.00 [bc] | 11.00 ± 0.00 | 20.80 ± 0.50 [de] | ≤0.05 | ≤1 |
| G 520 | 321.00 ± 37.00 [a] | 13.00 ± 1.00 | 21.80 ± 1.50 [de] | 0.03 ± 0.03 [bc] | ≤1 |
| CLI 70 | 317.33 ± 21.46 [a] | 12.00 ± 1.00 | 23.13 ± 4.39 [cde] | 0.02 ± 0.03 [bc] | ≤1 |
| CLI 275 | 232.00 ± 21.00 [bc] | 11.50 ± 0.50 | 24.55 ± 3.25 [bcd] | 0.07 ± 0.01 [ab] | ≤1 |
| CLI 1056 | 259.30 ± 37.07 [abc] | 10.50 ± 0.50 | 17.30 ± 2.88 [e] | 0.03 ± 0.03 [bc] | ≤1 |
| CLI 1109 | 248.50 ± 9.50 [bc] | 12.00 ± 0.00 | 21.90 ± 0.70 [de] | ≤0.05 | 3.60 ± 0.10 [a] |
| S-04 | 321.00 ± 21.00 [a] | 13.00 ± 0.00 | 16.5 ± 0.30 [e] | 0.11 ± 0.03 [a] | ≤1 |

Data are means ± standard deviations of three beer samples. Data with different superscript letters within each column are significantly different (Tukey tests: $p < 0.05$).

Lactic acid concentrations (317–321 ppm) are similar to commercial yeast S-04 in two strains (G 520, CLI 70), for the other strains, levels are between 208.5 and 278.5 ppm. The production of lactic acid by strain G 4 is noteworthy, since despite not having fermented all sugars available in the wort, it has produced levels almost comparable to other strains (G 502, CLI 275). Furthermore, yeast strain G 487 is the one that has produced the most lactic acid concentrations (417.50 ppm), which exceeds detection thresholds (400 ppm) [45].

Colour is one of the main characteristics of beer appearance. It is mainly determined by the amount of melanoidins produced in the malting process, the type of malt being used also very important. In this study, the recipe was always the same so that the differences observed are due to the yeast strains. Some studies have demonstrated that high fermentation yeasts produce beers with higher values of absorbance due to the browning and oxidation of the melanoidins [46].

Hops are mainly responsible for contributing to the bitterness of beer by providing α-acids, that will be transformed during boiling into iso-α-acids, which are the dominating bitter taste in beer [47]. The variety and quantity of hops was constant in all the beers; thus, it can be considered that the differences observed will be due to the influence of the strains studied. Some studies suggest that α-acid molecules of hops could adhere to yeast cell walls, which would settle to the bottom of the fermenter, thereby reducing the bitterness of the beer [48–50]. For the study, initial values were 32.77 ± 3.16 IBU, decreased for seven strains (G 450, G 502, G520, CLI 70, CLI 275, CLI 1056, CLI 1109) and only similar to S-04 for one strain (CLI 1056). In contrast, the levels remained stable for the strain that did not complete fermentation (G 4).

The presence of vicinal diketones (VDKs) is considered undesirable for beer quality. The most notable VDKs are diacetyl and 2-3 pentanedione which are important off-flavours of lager and ale beers. During fermentation, yeast cells excrete an intermediate of valine biosynthesis, α-acetolactate, that is, spontaneously decarboxylated to diacetyl. Both diacetyl and 2-3 pentanedione have a strong aroma of toffee and butterscotch with very low flavour thresholds, 0.15 ppm and 0.9 ppm respectively [51]. In this study, VKDs values are under threshold, therefore, no beer will contribute these undesirable aromas.

Sulphur dioxide ($SO_2$) have been added to foods for a long time, due to their numerous functionalities, as they can act as bleaching agents, antimicrobial agents, oxygen scavengers, reducing agents and enzyme inhibitors [52]. Yeasts, in addition to producing $H_2S$, can also produce small amounts of $SO_2$. Maximum levels of $SO_2$ in beer depends usually on how it is governed in different countries, in Spain it is 10 ppm. Only four yeast strains have produced it (G 354, G 450, G 487, CLI 1109) with 1.70–3.60 ppm levels, which are within the legally established levels, as well as below the threshold of perception (20 ppm) [53], above which they would produce unpleasant aromas. Furthermore, authors such as J. Dvořakand et al. and L. F. Guido [52,54], suggest that low $SO_2$ concentrations may contribute to the stabilisation of beer over time.

### 3.3. Aroma Compound Production in 1 L

Ethanol and carbon dioxide contribute to beer flavour, but the fermentation of wort generates a multitude of metabolites synthesised by yeast, which contribute to beer flavour. The production of the desired set of flavour-active compounds in beer is influenced by the choice of yeast strain and wort composition, as they depend on the metabolism of sugars and amino acids. The most important by-products of yeast metabolism (and related to beer quality) are sulphur compounds, organic and fatty acids, carbonyl compounds, higher alcohols and esters.

The experimental beers obtained were analysed for the content of the main higher alcohols, esters and acids (Table 3).

The higher alcohols or fusel alcohols are the most abundant organoleptic compounds in beer, with the highest concentrations in the fermenting wort at the time when free amino nitrogen drops to a minimum concentration. Concentrations below 300 mg $L^{-1}$ confer refreshing, flower and pleasant notes, and they impart desirable warming character compounds, which add complexity to the beer [29]. The total higher alcohols produced by the different strains do not exceed this concentration in any event (177.29 mg $L^{-1}$ in strain G 520) (Supplementary Materials: Table S1). In all cases studied here, the *Saccharomyces* strains showed similar or lower production of isobutanol, isoamyl alcohol, methionol and β-phenylethanol in comparison with commercial strain. Among these, G 520 strain produced the highest levels of these four higher alcohols and G 4 the lowest, except for methionol production, which are G 502 and CLI 70 respectively. The concentration of methionol in beer is generally below its odour threshold (2000 ppb) [55]. Furthermore, strains G 520, CLI 1056 and CLI 1109 produced isoamyl alcohol levels above the perception threshold (70 mg $L^{-1}$) [56], which imparts alcohol, banana and sweet flavours to beer. Ale strains, generally, produce more higher alcohols than lager strains, thus, the selection of an appropriate yeast strain can control their formation [57].

**Table 3.** Main volatile compounds found on ten selected yeast strains and S-04 in 1 L fermentation.

| Yeast Strains | G 4 | G 354 | G 450 | G 487 | G 502 | G 520 | CLI 70 | CLI 275 | CLI 1056 | CLI 1109 | S-04 |
|---|---|---|---|---|---|---|---|---|---|---|---|
| **Higher alcohols** | | | | | | | | | | | |
| Isobutanol | 3.70 ± 0.31 [e] | 15.29 ± 0.14 [b] | 10.77 ± 1.76 [cd] | 14.26 ± 0.74 [bc] | 7.58 ± 0.50 [de] | 25.71 ± 0.90 [a] | 14.42 ± 1.87 [bc] | 11.17 ± 0.46 [bcd] | 23.78 ± 1.82 [a] | 24.68 ± 1.59 [a] | 23.54 ± 2.78 [a] |
| Isoamyl alcohol | 29.77 ± 3.66 [f] | 69.38 ± 0.58 [b] | 53.39 ± 1.38 [de] | 62.07 ± 1.58 [bcd] | 51.53 ± 8.59 [e] | 97.73 ± 2.11 [a] | 58.32 ± 0.75 [cde] | 63.80 ± 2.15 [bcd] | 70.81 ± 1.93 [b] | 88.41 ± 2.45 [a] | 68.73 ± 5.45 [bc] |
| Methionol | 0.41 ± 0.02 [c] | 0.05 ± 0.02 [de] | 0.13 ± 0.00 [de] | 0.09 ± 0.03 [de] | 0.86 ± 0.01 [a] | 0.79 ± 0.03 [a] | 0.02 ± 0.00 [e] | 0.61 ± 0.06 [b] | 0.09 ± 0.01 [de] | 0.17 ± 0.11 [d] | 0.82 ± 0.09 [a] |
| β-phenylethanol | 2.98 ± 0.37 [i] | 9.98 ± 0.98 [h] | 11.57 ± 0.11 [gh] | 20.91 ± 0.57 [de] | 16.39 ± 0.67 [ef] | 52.99 ± 1.56 [a] | 11.70 ± 0.99 [fgh] | 23.07 ± 0.25 [d] | 37.46 ± 2.58 [c] | 16.10 ± 0.06 [fg] | 45.91 ± 4.09 [b] |
| **Esters** | | | | | | | | | | | |
| Isoamyl acetate | 0.22 ± 0.01 [e] | 0.61 ± 0.08 [de] | 0.42 ± 0.17 [e] | 0.40 ± 0.19 [e] | 1.30 ± 0.10 [c] | **2.34 ± 0.06** [a] | 0.89 ± 0.17 [cd] | 0.65 ± 0.04 [de] | 0.40 ± 0.01 [e] | **1.90 ± 0.32** [ab] | **1.81 ± 0.26** [b] |
| Ethyl hexanoate | 0.09 ± 0.02 [c] | 0.01 ± 0.01 [c] | 0.16 ± 0.00 [a] | 0.09 ± 0.07 [abc] | 0.03 ± 0.01 [c] | 0.02 ± 0.00 [c] | 0.07 ± 0.06 [abc] | 0.01 ± 0.00 [c] | 0.15 ± 0.01 [ab] | **0.60 ± 0.03** [bc] | 0.03 ± 0.00 [c] |
| Ethyl octanoate | 0.13 ± 0.02 [bcd] | 0.15 ± 0.00 [abcd] | 0.08 ± 0.07 [d] | 0.11 ± 0.00 [cd] | 0.21 ± 0.00 [a] | 0.18 ± 0.00 [ab] | 0.15 ± 0.01 [abcd] | 0.16 ± 0.01 [abc] | 0.09 ± 0.01 [cd] | 0.21 ± 0.02 [a] | 0.13 ± 0.00 [bcd] |
| 2-phenylethyl acetate | 0.01 ± 0.00 [d] | 0.02 ± 0.00 [d] | 0.02 ± 0.01 [bc] | 0.02 ± 0.00 [cd] | 0.03 ± 0.00 [ab] | 0.03 ± 0.00 [ab] | 0.03 ± 0.00 [ab] | 0.03 ± 0.00 [ab] | 0.03 ± 0.00 [ab] | 0.03 ± 0.00 [a] | 0.01 ± 0.00 [e] |
| **Fatty Acids** | | | | | | | | | | | |
| Isobutyric acid | nd | nd | 0.86 ± 0.01 [b] | 0.92 ± 0.02 [b] | nd | 0.88 ± 0.88 [b] | nd | 1.18 ± 0.00 [ab] | 1.72 ± 0.06 [a] | nd | nd |
| Butyric acid | nd | nd | 2.27 ± 0.15 [a] | 2.33 ± 0.04 [a] | nd | 2.35 ± 0.10 [b] | nd | 2.37 ± 0.05 [a] | 2.35 ± 0.02 [a] | nd | nd |
| Isovaleric acid | 3.24 ± 0.09 [f] | 6.09 ± 0.07 [bc] | 3.74 ± 0.06 [ef] | 3.81 ± 0.02 [ef] | 5.65 ± 0.17 [bcd] | 6.38 ± 0.13 [b] | 5.19 ± 0.04 [cd] | 5.95 ± 0.06 [bc] | 4.78 ± 0.21 [de] | 5.34 ± 1.07 [bcd] | 9.81 ± 0.59 [a] |
| Hexanoic acid | 0.81 ± 0.03 [e] | 2.31 ± 0.03 [bc] | 1.64 ± 0.14 [d] | nd | 2.57 ± 0.12 [b] | 2.68 ± 0.05 [ab] | 2.32 ± 0.44 [bc] | 1.90 ± 0.17 [cd] | 0.75 ± 0.01 [e] | 3.09 ± 0.22 [a] | 1.56 ± 0.07 [d] |
| Octanoic acid | 3.15 ± 0.18 [f] | 5.96 ± 0.01 [cd] | 5.71 ± 0.69 [d] | 4.18 ± 0.20 [ef] | 7.75 ± 0.19 [ab] | 8.63 ± 0.03 [a] | 7.15 ± 0.09 [bc] | 6.01 ± 0.36 [cd] | 4.22 ± 0.05 [ef] | 8.74 ± 0.67 [a] | 5.51 ± 0.35 [de] |
| Decanoic acid | 0.63 ± 0.06 [bc] | 0.38 ± 0.06 [bc] | 0.54 ± 0.16 [bc] | 0.45 ± 0.11 [bc] | 0.76 ± 0.40 [bc] | 2.57 ± 0.52 [a] | 0.42 ± 0.03 [bc] | 0.80 ± 0.00 [bc] | 0.55 ± 0.01 [bc] | 1.11 ± 0.74 [b] | 0.16 ± 0.02 [c] |
| **Guaiacol** | **0.06 ± 0.01** [b] | nd | **0.03 ± 0.03** [bc] | nd | **0.03 ± 0.03** [bc] | **0.05 ± 0.01** [b] | nd | nd | nd | **0.03 ± 0.03** [bc] | **0.13 ± 0.00** [a] |

Data, expressed as mg L$^{-1}$, are means ± standard deviations of three independent experiments. nd = not detected. Compounds above their threshold levels are marked in bold. Data with different superscript letters within each row are significantly different (Tukey tests: $p < 0.05$).

Esters are the most important group of flavour-active compounds synthesised by yeast during fermentation. Its industrial interest is due to the fact they have very low thresholds and define the fruity aroma of the beer. The results obtained showed important differences in levels of esters among the strains tested, G 520 being the strain which generated the highest content of total esters (3.06 mg $L^{-1}$). Isoamyl acetate imparts banana flavour with a threshold of 1.4 mg $L^{-1}$ [58], it was found in all beers, but only three of them overtake the threshold level (G 520, CLI 1109 and S-04). Yeast strain CLI 1109 was also remarkable in terms of ethyl hexanoate production, as it exceeds the threshold (0.21 mg $L^{-1}$) [59], providing a green apple aroma. Strain G 4, whose ethyl hexanoate levels did not exceed the threshold, and despite not having finished fermentation, showed higher concentrations than those produced by other strains. Therefore, it could also be used in co-fermentation with other *Saccharomyces* yeasts to enhance beer aroma [16].

Regarding fatty acids, Table 3 shows concentrations of hexanoic acid, octanoic acid and decanoic acid, which impart the so called caprylic flavour as an unpleasant flavour (thresholds values of 8, 15, 100 mg $L^{-1}$ respectively) [10]. This flavour is found in the majority of lager beers and in 20% of ale beers [60]. However, levels of these acids found in beer to contribute substantially to caprylic flavour are insufficient. All beers studied showed concentrations below threshold levels, with strain CLI 1109 producing the highest amounts of acids, except for decanoic acid where the strain G 520 was the most prominent. Other short chain fatty acids such as butyric, isobutyric and isovaleric acids were also studied. They usually increase during fermentation [61]. Butyric acid causes "cheesy" or "sickly" off-flavours and it was only detected in five strains (G 450, G 487, G 520, CLI 275 and CLI 1056). Isovaleric acid has an unpleasant aroma of old, rancid or sweaty cheese, making its presence in beer undesirable above a certain concentration. Isovaleric levels found varied between 3.24 and 6.38 mg $L^{-1}$ (9.81 mg $L^{-1}$ in commercial strain), possibly due to yeast metabolism, as they produce it from leucine [62]. Although both exceeded the perception threshold (butyric, 2 mg $L^{-1}$, isovaleric acid, 2.5 mg $L^{-1}$) [45], these aromas were not found in sensory analysis in the final beers. The combination of the parameters studied in beer and volatile compounds showed a correlation among alcohol and acids of r < 0.80, $p < 0.05$.

Wild *Saccharomyces* isolates tend to produce significant levels of phenolic off-flavours (POF), which give a clove-like aroma to beer. This could be considered as undesirable flavour in some beers, but POF aromas do not represent a problem for other styles, such as wheat beer, where clove-like aroma is part of the normal flavour profile [63]. In this case guaiacol with very low threshold (3.88 ppb) and smoky flavour [64] was studied. It was found in five of the ten beers studied (G 4, G 450, G 502, G 520 and CLI 110) as well as S-04 strain, which exceeded the threshold value.

The G 520 strain showed significant levels of all the by-products in comparison with the strain S-04 used as control.

### 3.4. Melatonin Production

Functional beers are defined as beers with health benefits for those who consume them moderately. In this case melatonin compound has a special mention, which is a sleep-regulating hormone that can modulate the circadian and seasonal rhythms, it has antioxidant properties and it can be produced in beer during alcoholic fermentation by yeasts [12]. Table 4 summarizes the data on ethanol and melatonin contents of the ten beers studied.

Levels found are between 5.04 and 56.51 ng $mL^{-1}$, whereas in two yeast strains it was not detected. Some previous studies suggest that melatonin production could be correlated with ethanol levels [12,65], whereas in this study it appears to have no direct correlation. For the yeast strains G 354 and CLI 275, melatonin was not detected, this could be attributed to its isolation origins and/or its use and therefore, to the different mechanisms of adaptation to the fermentation environment that different strains possess, as Vigentini et al. [66] and Morcillo-Parra et al. [67] described in their studies. The melatonin levels obtained are also

higher than in the above-mentioned studies with lager beers analysed with ELISA methods. In relation to other foods analysed by liquid chromatography (bread, tomato, black tea..), the concentrations found are higher than average [20].

**Table 4.** Impact of used yeast strain on melatonin and ethanol content in analysed beers.

| Yeast Strains | Melatonin (ng mL$^{-1}$) | Ethanol % (*v/v*) |
|---|---|---|
| G 4 | 33.60 ± 9.57 [b] | 1.59 ± 0.02 [e] |
| G 354 | nd | 5.07 ± 0.04 [b] |
| G 450 | 30.14 ± 0.09 [bc] | 4.63 ± 0.02 [c] |
| G 487 | 25.98 ± 5.52 [bc] | 4.55 ± 0.02 [c] |
| G 502 | 5.04 ± 0.95 [d] | 5.10 ± 0.04 [b] |
| G 520 | 28.87 ± 1.50 [bc] | 5.07 ± 0.05 [b] |
| CLI 70 | 31.61 ± 2.59 [bc] | 4.43 ± 0.14 [c] |
| CLI 275 | nd | 4.56 ± 0.01 [c] |
| CLI 1056 | 56.51 ± 1.66 [a] | 3.97 ± 0.16 [d] |
| CLI 1109 | 27.41 ± 1.86 [bc] | 5.07 ± 0.03 [b] |
| S-04 | 20.41 ± 5.25 [c] | 5.53 ± 0.05 [a] |

Data are means ± standard deviations. nd = not detected. Data with different superscript letters within each column are significantly different (Tukey tests: $p < 0.05$).

*3.5. Determination of Antioxidant Activity*

The antioxidant activity of experimental beers obtained by the different strains is reported in Table 5.

**Table 5.** Antioxidant activity of experimental beers.

| Yeast Strains | Q1 | Q2 | Qt |
|---|---|---|---|
| G 4 | 2.99 ± 0.20 [e] | 6.21 ± 0.37 [b] | 9.20 ± 0.57 [d] |
| G 354 | 3.89 ± 0.25 [abcd] | 9.14 ± 0.37 [a] | 13.03 ± 0.57 [abc] |
| G 450 | 3.93 ± 0.16 [bcde] | 8.60 ± 0.25 [a] | 11.64 ± 0.09 [abc] |
| G 487 | 3.39 ± 0.13 [de] | 8.25 ± 0.10 [a] | 11.64 ± 0.23 [c] |
| G 502 | 4.41 ± 0.28 [abc] | 9.04 ± 0.45 [a] | 13.48 ± 0.73 [ab] |
| G 520 | 4.74 ± 0.20 [a] | 8.68 ± 0.03 [a] | 13.43 ± 0.18 [ab] |
| CLI 70 | 3.84 ± 0.26 [cde] | 8.53 ± 0.37 [a] | 12.37 ± 0.24 [abc] |
| CLI 275 | 4.67 ± 0.59 [ab] | 8.80 ± 0.04 [a] | 13.67 ± 0.63 [a] |
| CLI 1056 | 4.31 ± 0.26 [abc] | 8.88 ± 0.47 [a] | 13.20 ± 0.63 [ab] |
| CLI 1109 | 4.07 ± 0.20 [abcd] | 8.19 ± 0.44 [a] | 12.26 ± 0.64 [bc] |
| S-04 | 3.78 ± 0.00 [cde] | 8.37 ± 0.21 [a] | 12.15 ± 0.21 [bc] |

Data are means ± standard deviations of three independent experiments expressed as millimoles of Trolox equivalents per litre (mmol TE L$^{-1}$). Q1, fast-acting antioxidants, Q2, slow-acting antioxidants, QT, total antioxidants. Data with different superscript letters within each column are significantly different (Tukey tests: $p < 0.05$).

The highest level of total antioxidant activity (Qt) was found in beers from G 502, G 520 and CLI 275 strains (13.43–13.67 mmol TE L$^{-1}$), whereas the lowest levels were found in beers from G 4 (9.50 mmol TE L$^{-1}$), followed by G 450 and G 487 (11.64 mmol TE L$^{-1}$). Strains G 502, G 520 and CLI 275 are also higher in Q1 values which are the fast-acting antioxidants that are oxidized in the first place. These can be considered more powerful than slow-acting antioxidants (Q2). Regarding other studies, antioxidant activity of the beers varied from 424.77 to 10,508.47 μmol TE L$^{-1}$ [13] which are lower values than the beers studied. It also depends on the methods used for the analysis, where can be found values between 3.70 and 29.11 mmol TE L$^{-1}$ analysed with ORAC method. This method determines the absorption capacity of oxygen radicals and measures the decreasing of fluorescence emission [68].

The combination of the parameters studied showed a correlation among antioxidant capacity and acids of r < 0.81, $p < 0.05$ and among antioxidant capacity and ethanol and glycerol of r < 0.77, $p < 0.05$.

### 3.6. Principal Components Analysis (PCA)

In the first instance, PCA was applied in order to evaluate the data trends. Two principal components (PCs) were extracted. PC1 explained up to 36.50% of the total variance and PC2 explained another 11.16%. By representing PC1 versus PC2, the scatter plot of the beers obtained by the strains with the selected parameters (biplot) was obtained (Figure 1).

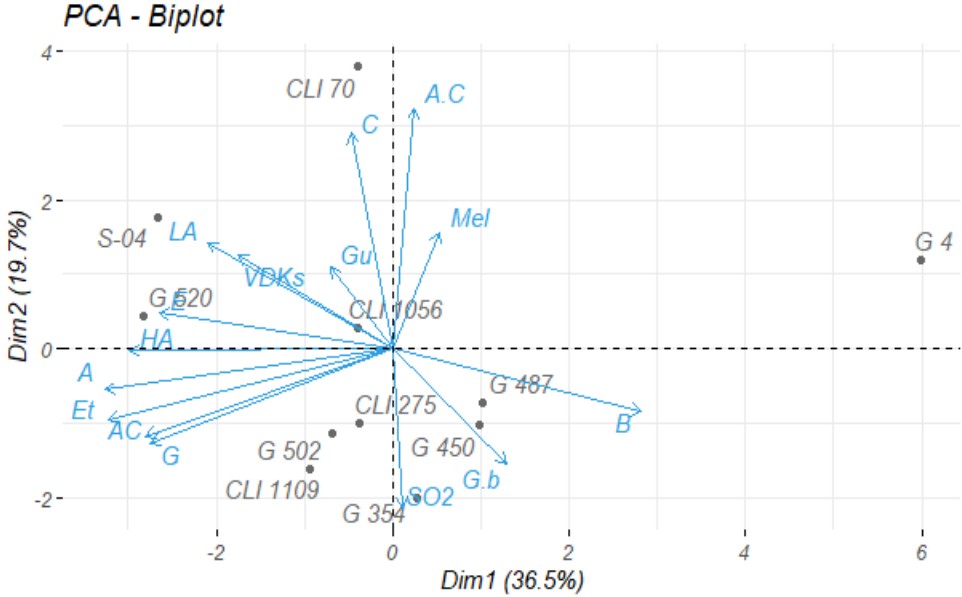

**Figure 1.** Projection of the beers on the axes formed by the principal components 1 and 2. Each object is the average of the three corresponding experimental beers. Mel, melatonin, B, bitterness, G.b, γ-butyrolactone, G, glycerol, AC, antioxidant capacity, Et, ethanol, A, total fatty acids, HA, total higher alcohols, E, total esters, LA, lactic acid, Gu, guaiacol, C, colour, A.C, total aldehydes/cetones.

Fermentations with G 487, G 450 and G 354 strains are situated in the positive region of PC 1, which is associated with bitterness, γ-butyrolactone aromatic compound and $SO_2$. On the negative side of PC1 and PC2 are projected the samples fermented with CLI 275, G 502 and CLI 1109, characterised by their high content of glycerol, antioxidant capacity, ethanol and total fatty acids. Strains G 520, CLI 70 and commercial strain (S-04) are grouped in the positive side of PCA 2 next to total higher alcohols, total esters, lactic acid, VDKs, guaiacol and colour. Finally, fermentation with strain CLI 1056 is situated in the centre of the two-dimensional spaces. Strain G 4 did not show a clear trend and did not cluster close to any group of compounds, probably influenced by its weak fermentation performance and the production of intermediate/low levels of all analysed parameters.

### 3.7. Sensory Analysis

The beers obtained by these fermentations underwent sensory analysis, with the data illustrated in Figure 2A–C. All the beers analysed showed significant differences for their main aromatic notes and some tasty flavours.

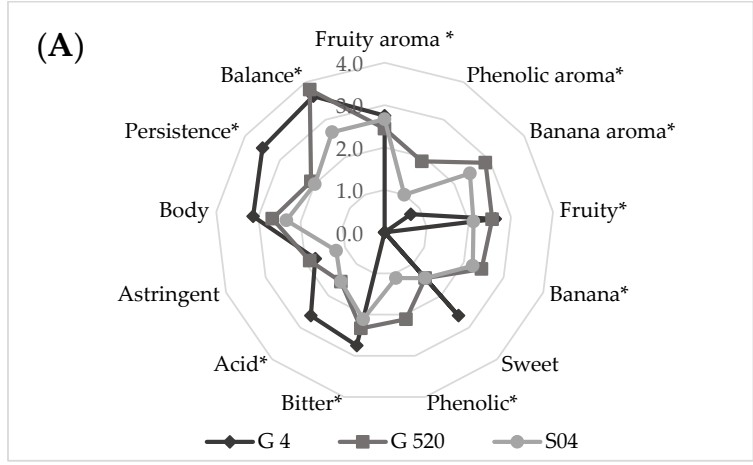

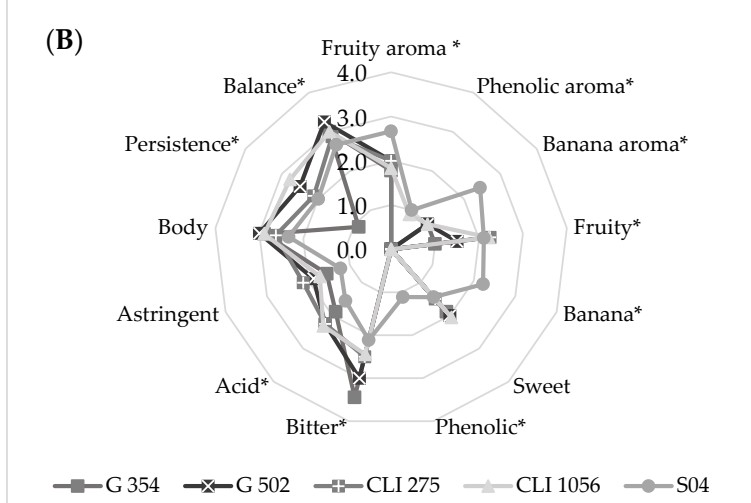

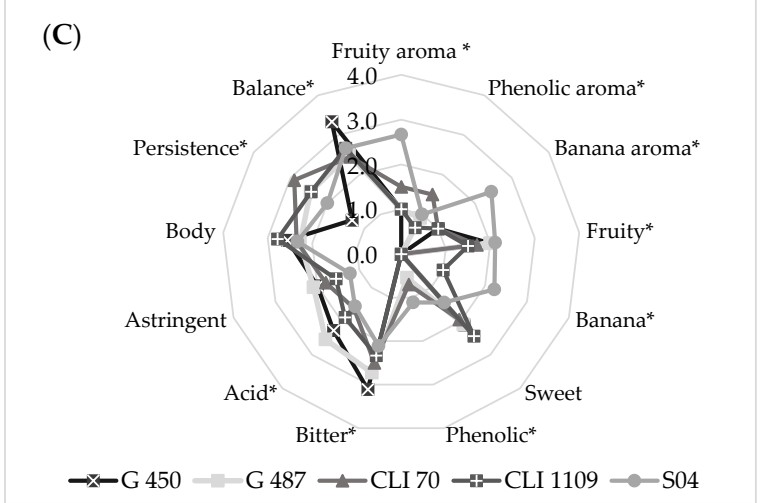

**Figure 2.** Sensory analysis of the beers studied. Aroma attributes: fruity aroma, phenolic aroma, banana aroma; flavour attributes: fruity, banana, phenolic, sweet, bitter, acid, astringent; overall attributes: body, persistence, balance. *, Significantly different (ANOVA; $p < 0.05$). (**A**), sensory analysis for strains G 4, G 520, S-04, (**B**) G 354, G 502, CLI 275, CLI 1056, S-04; (**C**), sensory analysis for strains G 450, G 487, CLI 70, CLI 1109, S-04.

Figure 2A represents those strains whose values in terms of aromas and fruity flavours were similar to the commercial strain. On the other hand, the rest of the parameters studied

exceed those of strain S-04, even for strain G 4, which did not complete fermentation and presented greater body and persistence. Those strains that showed slightly lower levels of fruity flavours are shown in Figure 2B, while those strains whose fruity aroma was half that perceived in the commercial strain are shown in Figure 2C. These strains generally showed higher values to strain S-04 in terms of bitterness, acidity, astringency, body, persistence and balance.

Strains G 487, G 520, CLI 70, CLI 1056 and CLI 1109 showed phenolic aromatic profile, that was not found in the other beers. The analyses showed a production of isoamyl acetate (banana flavour) with all the strains, but only three of them exceeded threshold values (G 520, CLI 1109, S-04), a fact that has been contrasted in tasting, as these are the strains in which this descriptor was detected. Isoamyl acetate levels are important for the perceived intensity of fruity aroma and are linked [69], being yeast strains G 520, CLI 1109 and S-04 the most highly rated in terms of fruity aroma.

The sensory bitterness of beer can be correlated with IBU values. In this sense, this relation was only shown by strain S-04, which had the lowest IBU values as well as the lowest tasting values, while the rest had no correlation. Some components in beer could contribute to or mask the bitterness such as sweetness due to sugars (residual sugar) that remain after the fermentation process. Thus, it is not possible to directly correlate IBU to the perceived sensory bitterness [70].

The sweetest beer, as expected, was the one fermented with the G 4 strain, because of the high residual maltose content. Many authors conclude that wort fermented by yeasts with low maltose fermentation abilities tend to increase the sensory sweetness of the beer [71]. Despite this, it was rated positively, with no worthy flavours, possibly masked by the fruity aromas and flavours found during tasting.

Acidity, body and astringency attributes were, however, identified as similar between all beers.

The results showed that the yeast strains were able to produce pleasant and aromatic sensory profiles with specific aroma characteristics, but beer fermented with the G 520 strain stands out from the rest in terms of aroma and fruity, banana and phenolic flavours, this was reflected in the analysis of the aroma compounds. Furthermore, this beer showed a good balance between aroma and flavours and medium persistence which makes it more palatable.

Beers with enhanced fruity characteristics and simultaneously reduced bitter attributes was reported to have a higher preference among consumers [72]. For this reason, and together with the results of previous analyses, strain G 520 was selected for industrial scale fermentation.

### 3.8. Industrial Scale Fermentation

Fermentations carried out at different scales could involve differences in the fermentation process with distinct kinetic behaviour and compound production and is therefore the main difficulty in studying new strains for industrial fermentations. Thus, scale-up studies that allow the identification of which parameters are dependent on the fermenter volume for the development of any fermentation process at industrial level are important [73].

At industrial scale, the rising $CO_2$ bubbles are responsible for the principal agitation in the system, inducing turbulence and leading to homogeneity in the substrate and cells. It must be considered that the agitation is not the same depending on the size of the fermenter. However, laboratory scale, with mechanical agitation, must achieve a better level of mixing [74].

In this sense, Table 6 shows a comparison of the main components analysed at different scales for the selected strain G 520 and the commercial strain S-04.

**Table 6.** Scale-up data comparison between 1 L fermentation and 100 L fermentation.

| Parameters | 1 L Fermentation | | 100 L Fermentation | |
|---|---|---|---|---|
| | **G 520** | **S-04** | **G 520** | **S-04** |
| Ethanol (% $v/v$) | $5.07 \pm 0.05$ [ab] | $5.53 \pm 0.05$ [a] | $4.15 \pm 0.23$ [c] | $4.60 \pm 0.43$ [bc] |
| Glycerol (g L$^{-1}$) | $3.19 \pm 0.14$ | $3.43 \pm 0.01$ | $3.11 \pm 0.18$ | $2.93 \pm 0.46$ |
| Lactic acid (ppm) | $321.00 \pm 37.00$ [ab] | $321.00 \pm 21.00$ [ab] | $268.50 \pm 6.36$ [b] | $337.00 \pm 17.00$ [a] |
| Colour (EBC) | $13 \pm 1.00$ [a] | $13 \pm 0.00$ [a] | $5.50 \pm 0.71$ [b] | $5.67 \pm 1.15$ [b] |
| Bitterness (IBU) | $21.80 \pm 1.50$ [a] | $16.50 \pm 0.30$ [b] | $22.35 \pm 0.49$ [a] | $23.40 \pm 3.33$ [a] |
| SO$_2$ (ppm) | $\leq 1$ [b] | $\leq 1$ [b] | $1.15 \pm 0.07$ [a] | $1.07 \pm 0.12$ [a] |
| Total higher alcohols (mg L$^{-1}$) | $177.29 \pm 4.60$ [a] | $139.08 \pm 12.41$ [b] | $91.11 \pm 5.21$ [c] | $80.41 \pm 5.20$ [c] |
| Total esters (mg L$^{-1}$) | $3.06 \pm 0.09$ [a] | $2.89 \pm 0.11$ [a] | $0.76 \pm 0.12$ [c] | $1.12 \pm 0.10$ [b] |
| Total fatty acids (mg L$^{-1}$) | $22.33 \pm 1.49$ [a] | $17.05 \pm 1.03$ [b] | $13.00 \pm 0.18$ [c] | $17.01 \pm 0.66$ [b] |
| $\gamma$-Butyrolactone (mg L$^{-1}$) | $0.25 \pm 0.00$ [a] | $0.27 \pm 0.00$ [a] | $0.13 \pm 0.02$ [b] | $0.15 \pm 0.04$ [b] |
| Guaiacol (mg L$^{-1}$) | $0.05 \pm 0.01$ [b] | $0.13 \pm 0.00$ [a] | nd | $0.07 \pm 0.02$ [b] |
| Melatonin (ng mL$^{-1}$) | $28.87 \pm 2.13$ | $20.41 \pm 5.25$ | $22.04 \pm 3.33$ | $22.87 \pm 3.07$ |
| Antioxidant capacity (Qt) (mmol TE L$^{-1}$) | $13.43 \pm 0.18$ [a] | $12.15 \pm 0.21$ [ab] | $12.06 \pm 2.15$ [ab] | $10.05 \pm 0.52$ [b] |

Data are means $\pm$ standard deviations of three independent experiments. nd = not detected. Data with different superscript letters within each row are significantly different (Tukey tests: $p < 0.05$).

Starting with the fermentation kinetics, it was observed that in the 100 L fermentation without stirring, it was prolonged in time up to 2 weeks, while in 1 L with agitation it was approximately one week. Numerous studies have shown variations in fermentation kinetics in both shaken and unstirred fermentations, because of two reasons: substrates and cells are not equally distributed, and there are low mass transfer volumetric coefficients [74,75]. Furthermore, refermentation occurring during bottle conditioning also plays an influential role in terms of ethanol production, as it can contribute to an increase in ethanol production up to 0.5% $v/v$ [76]. In consequence, keeping all other parameters constant (temperature, yeast starter...) and taking into account refermentation in 1 L as well as agitation and scale up, ethanol production was reduced in both strains in 100 L beers, as reported in other studies [73,76]. Glycerol production was less noticeable, with only the commercial strain showing differences.

The bitterness concentrations found in 100 L are higher than in 1 L for both strains, this may be because the cell biomass produced during fermentation is lower than that produced in 1 L as a consequence of the agitation, thus avoiding the adherence of the $\alpha$-acids to the cell wall. Therefore, further in-depth studies would be necessary to determine this.

Differences in colour are due to the Maillard reactions that occur during wort boiling [77], as the wort used in the 1 L fermentations was sterilised prior to use, thus causing a darkening of the wort, which did not occur in the 100 L beer as it was transferred directly to the fermenter after the wort had cooled down.

Lactic acid and SO$_2$ production remain constant when considering the detection ranges of the FoodLab equipment and the standard deviations in G 520 and S-04.

Regarding the volatile compounds formation and as previously discussed, both agitation, scaling and refermentation not only influenced ethanol levels, but also influenced the production of volatiles [78]. Their reduction was very noticeable in terms of higher alcohols and consequently, in the reduction of esters production, as they are dependent on alcohols [34]. This could be observed especially in the reduction of the concentration of isoamyl alcohol and consequently of isoamyl acetate.

Finally, antioxidant capacity and melatonin production showed up not being dependent on scaling and stirring for both strains.

### 3.9. Consumers

Consumer acceptability is the final stage to determine if a testing beer is suitable for full scale production in the breweries, thus identifying the reason for the success (or failure) of a product and its market opportunities [79].

To test the beer fermented in 100 L with G 520 yeast strain, 112 volunteer consumers were recruited. The prerequisites for participating in the study were that the individual habitually consumed beer. The panel was composed of 63.72% females and 36.28% males, with an average consumer age of 44.83 ± 14.54 years old and ranged from 20 to 73 years old. Regarding the frequency of beer consumption, 97.27% of the panel reported frequently drinking beer every day or during the weekend. A total 70% consumed both commercial and craft beer, whereas 30% only consumed lager beer.

Beer appearance was evaluated regarding foam consistency and visual impression. Consumers rated the consistency of the foam as light-fine, while visual impression was rated mainly as hazy (Table 7).

**Table 7.** Analysis of the data frequency obtained by CATA questions.

| Sensory Attributes | Response (*n* = 112) |
|---|---|
| **Appearance** | |
| Foam consistency | |
| Light | 54 |
| Fine | 29 |
| Medium | 21 |
| Persistent | 4 |
| Creamy | 4 |
| Visual impression | |
| Very haze | 19 |
| Hazy | 49 |
| Dull | 10 |
| Clear | 24 |
| Bright | 10 |
| **Aroma** | |
| Aroma intensity | |
| Low | 3 |
| Low-medium | 13 |
| Medium | 31 |
| Medium-high | 51 |
| High | 14 |
| **Taste** | |
| Acidity | |
| Low | 39 |
| Low-medium | 42 |
| Medium | 20 |
| Medium-high | 9 |
| High | 2 |
| Bitterness | |
| Low | 35 |
| Low-medium | 38 |
| Medium | 29 |
| Medium-high | 8 |
| High | 2 |
| Mouthfeel body | |
| Light | 9 |
| Light-medium | 34 |
| Medium | 53 |
| Medium-full | 12 |
| Full | 4 |
| Aftertaste intensity | |
| Short | 0 |
| Short-medium | 6 |
| Medium | 30 |
| Medium-long | 50 |
| Long | 26 |

For aroma attributes (Figure 3), malty, hops, fruit, banana and phenolic were the most frequent aromas found in beer, with ranges between 36.61 and 59.82%.

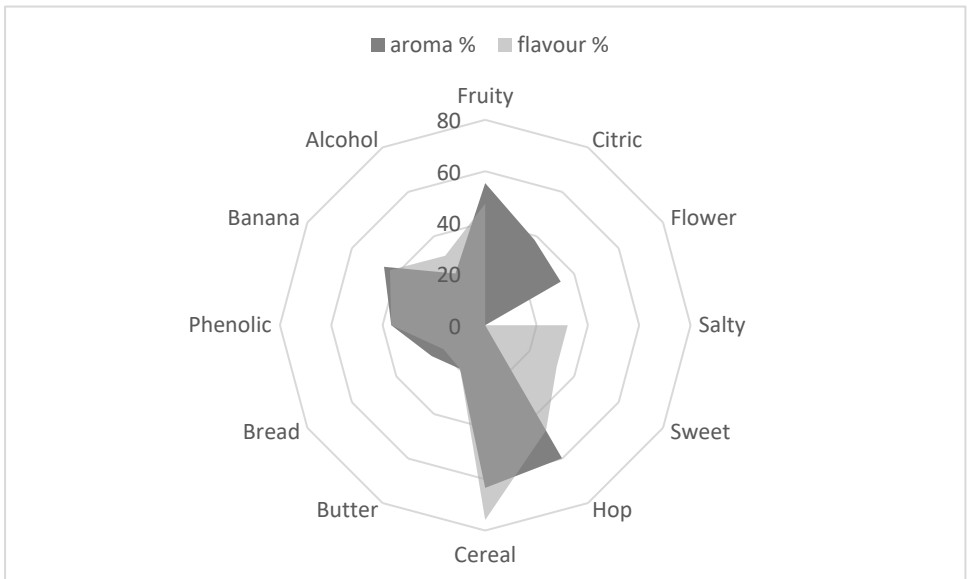

**Figure 3.** Percentage of detection by consumers for each of the analysed attributes in aroma and flavour.

Regarding flavour attributes, malty, fruity, hop, banana and phenolic were found as the most predominant. Considering the previous analyses as well as the sensory evaluation by the trained tasting panel, fruit, banana and phenolic aromas and flavours continued to predominate in the beer. In addition, consumers also highlighted the malty and hoppy flavours. Some previous studies suggest that consumers have a preference for conventional flavours, cereals, yeast and hops as the most preferred [80].

Degree of bitterness and acidity were also important factors, with low-medium levels found by the 85.00% of consumers and 67.50%, respectively.

Individual attributes, which may be considered good or bad, may have a different place in the overall perception of taste. For the tasted beer, the overall impression (from 1 to 9) was 6.59 points for 71.43% of consumers, with a purchase intention of a 66.96% for the total of consumers.

## 4. Conclusions

This study presents the ability of wine *Saccharomyces* yeast strains for beer fermentation. Although the original wort was similar in all fermentation experiments, the beers produced showed markedly different fermentation performances and sensory profiles. Not all strains were able to ferment maltose, however strain G 4 was selected for its production of aroma compounds as well as its balance and melatonin production. It could therefore be used to produce low ethanol beer or as bioflavouring agent in co-culture fermentations with standard brewing yeast strains, and additionally obtain potentially functional beers. Further studies will focus on the development of these beers. In relation to the rest of the yeast strains selected and analysed, they stood out for their fruity aroma and flavour, balanced acidity and bitterness, as well as a marked balance between aroma and flavour. All of this was aimed at producing a beer that would fit in with the style desired by the La Cibeles brewery, fruity flavour, without undesired aromas and with an adequate fermentation capacity. It could also be used for the production of Weizen beer styles due to its fruity character, particularly marked in terms of banana, as well as spicy aromas of clove.

We found that ester content and moderate levels of bitterness determine the preference of beer fermented with G 520 strain by the panel and for consumers. In addition, beers

showed a melatonin production and antioxidant activity above the average of other foods, which is necessary for the development of functional foods.

**Supplementary Materials:** The following are available online at https://www.mdpi.com/article/10.3390/fermentation7040290/s1, Table S1: Volatile compounds produced by the ten different *Saccharomyces* strains in 1 L fermentation.

**Author Contributions:** Conceptualization, J.M.C. and T.A.; Funding acquisition, T.A.; Investigation, V.P., M.G. and T.A.; Methodology, V.P. and T.A.; Resources, T.A.; Supervision, T.A.; Writing—original draft, V.P.; Writing—review & editing, V.P. All authors have read and agreed to the published version of the manuscript.

**Funding:** This research was funded by the DG Research and Technological Innovation of the Regional Government of Madrid, Spain, grant number IND2017/BIO7787 for the industrial doctorate carried out by Vanesa Postigo. The project was developed between La Cibeles brewery and the Madrid Institute for Rural, Food and Agriculture Research and Development (IMIDRA) (Madrid, Spain).

**Informed Consent Statement:** Informed consent was obtained from all subjects involved in the study.

**Conflicts of Interest:** The authors declare no conflict of interest.

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
