# Peer review of "Wine Saccharomyces Yeasts for Beer Fermentation"

_fermentation, doi:10.3390/fermentation7040290_

Round 1

Reviewer 1 Report

Dear Authors,

Manuscript ID: fermentation-1477518

  This study aimed  to evaluate the ability of  Saccharomyces yeast strains isolated from the Madrilenian agriculture to produce a novel  beer. Research is very interesting as well as has a scientific value.

The introduction provides a good, generalized background of the topic that quickly gives the reader   appreciation of the scientific relevance and timeliness of the research theme. The advantage of this manuscript is the very carefully described research material and a rich set of applied experimental methods.  

I think that the findings of this study properly described in the context of the published literature. The conclusions were supported by an appropriate evidence. 

However, there are flaws of the manuscript that  need to be fixed before publication.

In my opinion, the main drawback of this manuscript is presentation of the results of sensory evaluation of developed products. This part of the study needs to be improved necessarily.

 Specific comments on the manuscript are as follows:

  1. Line 33: Please, replace the word “numerous” with another – “several”.
  2. Line 35: Please, give more examples of references.
  3. Please, in the subsection no. 1,  explain the term “ Ale beer”. A reader needs to know to which type of fermentation the product is associated with ?
  4. Line 54: Please, replace the word “functional” with “potentially functional”. There was no  performance evaluation of the developed products in human clinical trials.
  5. Lines 120-130: Please, complete reference.
  6. Line 131: Please, change the numeration of subsection to 2.3.2. Please, proceed in the same way for the following subsections of the manuscript.
  7. Line 181: Please, complete information about PCA analysis and justify an application of this method.
  8. The subsection Statistical Analysis, should be placed at the of the section 2 Materials and Methods.
  9. Line 184: subsection no. 2.5 Sensory Analysis: Please, complete the standard by which the sensory members  were trained. What method of sensory analysis was used for estimation of sensory quality of developed products? Please, also complete reference.
  10. Line 193: (subsection 2.6. Consumer acceptability test): Please, complete the customers characteristics (gender, age).
  11. Line 193: (subsection 2.6 Consumer acceptability test) Please, complete reference.
  12. Line 246; 274; 353: (the titles of Tables 1-3): Please, detail the term of  “strains” by adding the word “yeast” in front of the phrase.
  13. Tables 4-5: Please, complete term “yeast” in the first column of each table.
  14. Line 391: Please, change the title of Table 4 to : “Impact of used yeast strain on melatonin and ethanol content in analysed beers”
  15. Tables 4-6: Please, complete the results of statistical tests.
  16. Lines 426-431: This information must be included in the subsection Statistical Analysis.
  17. Line 571: (subsection 5 Conclusions) Please, give the results of presented study to the application in practice.

From my standpoint, this manuscript is appropriate for publication in Journal – Fermentation after minor revision, given the above aspects.  

Reviewer 2 Report

The manuscript "Wine Saccharomyces Yeasts for Beer Fermentation" by Postigo et al. is an interesting search for new brewing yeasts. Exploring the natural diversity for new industrial yeast strains is a strong trend in food microbiology. The approach is not novel, but the experiments here are very systematic, with a big number of yeasts screened and a particular yeasts analysed in big volumes, so the whole process has been covered and a new brewing strain has been described and ready to use. However, there is not indication of the flocculation properties of this strain or other isolated. Serial repitching is important to the industrial performance of any commercial yeast, so some indication of such process should be included.

Minor points.

Line 14. It seems all yeast come from vineyard/wine environments. That should be  clearly indicated, it is not a screen in many agricultural environments.

Line 353. Table 3. It would be nice to indicate the volatile products that are above the expected perception threshold, by shadowing those cells for instance.

Reviewer 3 Report

In this article, authors explore the capacities of a wide collection of yeast strain belonging to the S. cerevisae species to ferment wort for making beer. It is a very complete study, including fermenting compounds, volatile compounds, melatonin production and antioxidant capacity as well as sensory analysis and consumer acceptancy test. They have performed, no just lab scale trials,  seven mall size brewery scale fermentation.

From the whole collection they extract ten (nine from my point of view) as very suitable for beermaking and, among them, one with good capacities. As a result of this work, it would be possible to apply this strain to industrial conditions of brewery.

Conclusions are consistent and they address the main objective of the paper. The manuscript is also well written and presented, so it is perfectly suitable for publication in "Fermentation".

Despite this, you can find below more minor comments.

  • Line 206: To achieve this objective, it was necessary to study the abilities to ferment maltose, the production level of H2S and their fermentative behavior n wort. I suggest separating these three results for better comprehension of the text.
  • Line 220: A 70.9% of the strains 220 showed a moderate production of H2S (Type II), whereas a 10.3% showed a high production (Type III) and a 18.8% a low-null production. Production of SH” is a very problematic issue in  I wonder Type III why was chosen. Also I would be interesting to know which type is trainS04.
  • Line 254: G 4 was not able to ferment maltose. In my opinion this strain not suitable for making beer.
  • Line 267: Glycerol levels: would be interesting add to table 1
  • Line 299: Bitterness decreased. Some explanation is required (IBU is a sensorial index, more sugar, less bitterness)
  • Line 313: It is a by-product desirable in beer as it acts as an 313 antioxidant and stabilizes the reactive carbonyl components that are responsible for off flavors in matured beer. SO2 produced by a yeast is combines and can not act as antioxidant or stabilizing agent. Please, avoid this comment. In fact, it is desirable yeast strains that do not produce SO2
  • Table 3: Include summatory of higher alcohols, Esters, Acids
  • Table 3: fatty in Fatty Acids is missing.
  • Table 5: Indicate explanation of Q1, Q2 and Qt, as in the text.
  • Line 576: strain G4: It could therefore be used to produce low ethanol beer: This strain that cannot ferment maltose and cannot produce beer and their use should bring a lot of stabilization problems in bottles. I wouldn’t be a low ethanol beer, because It wouldn’t be a beer really.
